# Modulation of sHLA-G, PD-1, and PD-L1 Expression in Cervical Lesions Following Imiquimod Treatment and Its Association with Treatment Success

**DOI:** 10.3390/cancers16071272

**Published:** 2024-03-25

**Authors:** Andrej Cokan, Neila Caroline Henrique da Silva, Rajko Kavalar, Igor But, Maja Pakiž, Sheilla Andrade de Oliveira, Fabiana Oliveira dos Santos Gomes, Rodrigo Soares da Silva, Christina Alves Peixoto, Norma Lucena-Silva

**Affiliations:** 1Department for Gynaecological and Breast Oncology, University Medical Centre Maribor, 2000 Maribor, Slovenia; majapakiz@gmail.com; 2Laboratório de Imunogenética, do Departamento de Imunologia, Instituto Aggeu Magalhães-Fiocruz, Campus da UFPE, Recife 50740-465, Brazil; neila_carol0212@hotmail.com (N.C.H.d.S.); sheilla.andrade@fiocruz.br (S.A.d.O.); norma.silva@fiocruz.br (N.L.-S.); 3Department for Pathology, University Medical Centre Maribor, 2000 Maribor, Slovenia; rajko.kavalar@gmail.com; 4Department for General Gynaecology and Gynaecological Urology, University Medical Centre Maribor, 2000 Maribor, Slovenia; igor.but@ukc-mb.si; 5Laboratório de Ultraestrutura, do Departamento de Entomologia, Instituto Aggeu Magalhães-Fiocruz, Campus da UFPE, Recife 50740-465, Brazil; fabiana.gomes@unicap.br (F.O.d.S.G.); rodrigosoares321@gmail.com (R.S.d.S.); christina.peitoxo@fiocruz.br (C.A.P.)

**Keywords:** imiquimod, cervical intraepithelial lesion, PD-1, PD-L1, HLA-G, immune checkpoint inhibitors

## Abstract

**Simple Summary:**

This study investigates the therapeutic potential of imiquimod (IMQ), a synthetic toll-like receptor 7 agonist, for treating cervical intraepithelial neoplasia (CIN) associated with human papillomavirus (HPV) infection. Given the risks associated with conventional surgical treatments, IMQ non-invasive application makes it an attractive alternative. This study specifically explores the correlation between IMQ treatment and the expression levels of immune checkpoint molecules PD-1, PD-L1, and sHLA-G in cervical lesions. The results suggest that baseline sHLA-G levels may predict treatment outcomes, while PD-L1 expression before treatment correlates with success. Understanding these immunomodulatory effects sheds light on IMQ potential as a conservative treatment for high-risk cervical lesions.

**Abstract:**

(1) Background: Cervical intraepithelial neoplasia (CIN) is a precancerous condition linked to human papillomavirus (HPV) infection, often necessitating surgical interventions carrying the risk of subsequent preterm births. This study explores the potential of imiquimod (IMQ), as a non-invasive alternative treatment. The focus is on understanding IMQ impact on immune checkpoint molecules, particularly PD-1, PD-L1, and sHLA-G, which play pivotal roles in shaping immune responses and cancer progression. (2) Methods: Forty-three patients diagnosed with a high-risk squamous intraepithelial lesion (HSIL, p16-positive) self-applied 5% IMQ encapsulated in sachets containing 250 g of cream into the vaginal cavity three times a week for 16 weeks. The impact of IMQ therapy on cervical lesion regression was assessed through immunohistochemistry (IHC), examining changes in sHLA-G, PD-L1, and PD-1 levels. The antiviral activity of IMQ was evaluated through HPV-E7 immunofluorescence. Ethical considerations were adhered to, and the research methods were based on a previously approved clinical trial (clinicaltrials.gov Identifier: NCT04859361). (3) Results: IMQ treatment demonstrated efficacy, leading to lesion regression. sHLA-G levels in CIN before starting IMQ application were associated with unsuccessful treatment (*p* = 0.0036). IMQ did not significantly alter the expression of PD-1. We observed a decrease in PD-L1 levels in those who were successfully treated (*p* = 0.0509) and a reduction in HPV burden. (4) Conclusions: IMQ exhibits promise as a non-invasive treatment for CIN, emphasising its potential to modulate the immune microenvironment. Baseline sHLA-G levels emerge as potential predictors of treatment response. Understanding the nuanced dynamics of immune checkpoints sheds light on IMQ mechanism of action. Further exploration is warranted to decipher the intricate mechanisms underlying IMQ treatment in the context of cervical lesions.

## 1. Introduction

Cervical intraepithelial neoplasia (CIN) is a precancerous condition that is associated with human papillomavirus (HPV) infection [1]. Standard treatment for a high-risk squamous intraepithelial lesion (HSIL), which includes CIN2 and CIN3, is surgical removal of the lesion, most commonly with LLETZ, but surgery increases the risk of preterm births in subsequent pregnancies [2]. For this reason, alternative conservative methods for SIL treatment are constantly being evaluated, and imiquimod (IMQ), a synthetic toll-like receptor 7 (TLR7) agonist, has generated attention for its potential to stimulate local antitumor and antiviral immune responses. Although studies have shown that IMQ is effective for treating vulvar intraepithelial neoplasia (VIN) [3], vulvar Paget disease [4], vaginal intraepithelial neoplasia (VaIN) [5], and CIN when compared to placebo or no intervention [6], its exact mechanism of action remains unclear [7].

Immunomodulation is emerging as a key determinant of cancer progression and treatment response. The interactions between tumour cells in the immune microenvironment play a crucial role in shaping disease outcomes. Immune checkpoint molecules, such as PD-1 and PD-L1, are central players in maintaining immune homeostasis in regulating the extent of immune responses. Notably, PD-1 expression on T cells and its interaction with PD-L1 expressed on tumour cells have been linked to immune evasion and tumour growth in cervical cancer [8]. Furthermore, sHLA-G, a non-classical human leukocyte antigen class I molecule, has also emerged as a potential immunomodulatory factor in cancer. Elevated levels of sHLA-G have been associated with immune tolerance and cancer progression, and its expression in the tumour microenvironment may contribute to the suppression of antitumor immune responses [9]. Although we have data demonstrating that sHLA-G expression in cervical cancer lesions is associated with cancer prognosis [10], the specific role of sHLA-G in the context of treatment with IMQ remains to be elucidated.

In this study, we aimed to investigate the potential correlations and modulations between PD-1, PD-L1, and sHLA-G expression levels in response to the treatment of cervical lesions with IMQ and their association with treatment outcomes. By doing this, we also aimed to uncover the mechanisms by which IMQ influences these pathways and its potential implications for cervical lesion management. Understanding these immunomodulatory effects could shed light on IMQ role as a non-invasive treatment option for HSILs, offering a promising alternative to traditional surgical interventions.

## 2. Materials and Methods

### 2.1. Patients and Study Design

In a previous investigational study (ClinicalTrials.gov Identifier: NCT04859361), our research findings substantiated the efficacy of topical IMQ treatment in facilitating the regression of cervical lesions [11]. The patients enrolled in the trial were sourced from organised screening and were not immunodeficient, immunocompromised, or undergoing corticosteroid treatment. Within the cohort of 52 patients who underwent IMQ therapy, a total of 43 (82.7%) participants adhered to the prescribed treatment regimen. Data regarding baseline characteristics, including age, smoking history, and histopathological diagnosis, were collected in the previous study [11]. The IMQ formulation employed in this study comprised a 5% concentration, encapsulated in sachets containing 250 g of cream. The administration protocol necessitated the self-application of the IMQ cream into the vaginal cavity using a menstrual cup, with an application duration spanning 6 to 8 h. This application frequency was set at three times per week, for an overall treatment duration of 16 weeks, representing the maximum allowable therapeutic duration.

In instances where patients experienced severe and intolerable side effects attributable to IMQ treatment, a predefined protocol allowed for a reduction in the frequency of applications. This reduction was implemented initially as biweekly application and, if side effects persisted, further decreased to a once-weekly regimen to mitigate discomfort and enhance treatment tolerability. 

To comprehensively assess the effectiveness of IMQ therapy, an evaluation schedule was adopted. Specifically, punch biopsies were conducted at the 10-week mark to proactively exclude disease progression. Subsequently, at the 20-week mark following the initiation of treatment, identical procedures were performed to assess the overall success of the therapeutic intervention. During this 20-week evaluation, biopsy specimens were obtained from sites where lesions had been previously identified. Additionally, supplementary biopsies were undertaken in the event of newly detected lesions to ensure a thorough and comprehensive evaluation of the treatment’s impact.

Following collection, all biopsy specimens were transported for IHC analyses to the Laboratório de Imunogenética within the Department of Immunology and the Laboratório de Ultraestrutura within the Department of Entomology, both situated at the Instituto Aggeu Magalhães-Fiocruz in Recife, Brazil. All the data used in this study were collected during our previous study [11].

Our previous trial received ethical approval from the National Medical Ethics Committee of the Ministry of Health under reference numbers 0120-13/2017 and KME 58/02/17 (approval date 10 May 2017). The original study protocol explicitly included provisions allowing for additional analyses to be conducted on tissue samples without necessitating further ethics committee approval. This adherence to established ethical guidelines and prior approval ensures that all research activities, including the subsequent analyses on tissue samples, align with ethical standards.

### 2.2. IHC Evaluation

Sections of 5 µm thickness from cervical biopsies were submitted to IHC reactions using a Dako EnVision™ FLEX+ kit (Dako, Code: K8002, Santa Clara, CA, USA), following the manufacturer’s instructions. The primary antibodies were mouse anti-PD-1 antibody (ab52587; [NAT105] Abcam, Cambridge, UK), rabbit anti-PD-L1 monoclonal (ab213524; [EPR19759] Abcam), and mouse anti-soluble HLA-G (sHLA-G) isoforms 5 and 6 (5A6G7-Exbio, Vestec, Czech Republic). Protein expression was visualised and recorded using a microscope (Observer Z1, Zeiss MicroImaging, GmbH, Jena, Germany) equipped with a camera and image analysis software 2.1.0 (AxionCam MRm Zeiss). Brownish-brown staining revealed positive samples. Each slice was photographed in four fields. Three independent experts determined the pixel value acquired from the images using the Gimp 2.10.18 program (GNU Image Manipulation Program, UNIX platforms, www.gimp.org (accessed on 5 September 2023) and the mean values used in the analysis of protein expression in response to topical treatment with IMQ.

### 2.3. Immunofluorescence

Cervical biopsy samples were embedded in paraffin, and sections (5 µm) were cut using an RM 2035 microtome (Reichert S, Leica, Wetzlar, Germany), mounted on glass slides, rehydrated, and treated with 20 mM citrate buffer (pH 6.0) at 100 °C for 30 min. They were then permeabilised with 0.5% Triton X-100 and blocked for 1 h with 3% BSA plus 0.2% Tween 20 in Tris-buffered saline. The cervical slides were then incubated with Anti-HPV11 E7 [9H5] antibody (Abcam, catalogue number ab100967) and diluted 1:100. Incubation with the primary antibody was performed overnight. Then, the slides were incubated with a secondary monoclonal antibody conjugated to Alexa fluor 488 (Abcam, catalogue number ab150113) against mouse immunoglobulin for 1 h at room temperature, and, subsequently, DAPI was added at a concentration of 1:500 for 5 min. The slides were washed, mounted in Prolong Gold Antifade fluorescent medium (Life 61 Technologies, catalogue number: P36930), observed under a Leica DMI8 fluorescence microscope, and processed with the Leica Application software 1.4.6. 28433 LAS Suite (Leica Microsystems, Wetzlar, Germany). The figures were exported as tiff files with Adobe Photoshop version 8.

### 2.4. Statistical Analysis

The Shapiro–Wilk test was applied to test the assumption of normality of the variables involved in the study. The variables with values of *p* < 0.025 were selected, ordered from the lowest to the highest significance value, and inserted into the model, leaving only the variables that presented a significance of *p* < 0.05. The differences in means for the independent variables were evaluated using Student’s *t* test when assumptions of normality were observed; when they were not observed, the Mann–Whitney test was used to assess the medians of the variables. In the paired analysis approach involving two moments over time, the paired Student’s T test was used when normality was observed in the variables, and when normality was not observed, the Wilcoxon test was used. Logistic regression was used to perform a univariate analysis and a subsequent multivariate analysis. The magnitude of these associations was estimated with an Odds Ratio (OR) using 95% confidence intervals (CIs). OR values were adjusted using multivariate logistic regression according to possible interaction variables. All conclusions were considered at a 5% significance level. R Core Team software (2023) was used in evaluating the results of the study [12].

## 3. Results

### 3.1. Characterisation of the Cervical Lesion Regression and Vaginal Inflammation Reaction after IMQ Therapy

Forty-three (82.7%) patients who completed the treatment regimen with IMQ were included in this study. The patients who underwent topical treatment with IMQ were on average 28.4 years old, with 21 having a starting diagnosis of CIN2 and the remaining CIN3. The patients with CIN2 were 5 times more likely to be successfully treated (63% × 37%; CI (95%) = 1.4–22.5; OR = 5.1; *p* = 0.0202). Treatment success was not associated with the number of IMQ applications (*p* = 0.8016), patient age (*p* = 0.3100), or smoking status (*p* = 0.9104). 

The occurrence of side effects has been previously detailed in our earlier study. These included local effects such as vaginal or vulvar inflammation and lower urinary tract symptoms, as well as systemic symptoms like fatigue, malaise, and fever, reported in 88.5% (46/52) of the patients using IMQ. Moderate (grade 2) and severe (grade 3) side effects were present in 38.5% (20/52) and 13.5% (7/52) of the patients, respectively. Nine patients who discontinued IMQ use, mainly due to severe side effects, were excluded from this study. The protocol, involving applying IMQ at least twice per week for 16 weeks, was strictly adhered to by 81.4% (35/43) of the women, and treatment was successful in 60.0% (21/35) of cases. Only 18.6% (8/43) of the patients used IMQ irregularly or temporarily stopped treatment. However, even within this subgroup, the success rate was notably high, with treatment achieving success in 62.5% (5/8) of the patients. Vaginal inflammation, assessed through a clinical examination with or without a bacterial swab after 10 weeks of treatment with IMQ, occurred in a similar proportion among the women who exhibited lesion regression (41% of 27) and those who did not (50% of 16) at the completion of the treatment protocol (*p* = 0.7846). However, after the completion of topical treatment, the group of women who responded to the IMQ therapy showed a higher frequency of vaginal inflammatory reactions (37% × 19%; OR = 7.4; CI (95%) = 1.7–32.3; *p* = 0.0103).

A visual analysis of IHC staining in cervical samples, collected before and after IMQ treatment, indicated a potential relationship between changes in sHLA-G and PD-L1 and possibly PD-1 levels with treatment success (Figure 1). 

### 3.2. Effect of IMQ Treatment on Cervical sHLA-G Levels

Elevated sHLA-G levels in HSILs before initiating IMQ application were associated with unsuccessful treatment (*p* = 0.0036) (Figure 2A). However, there was no difference in the tissue levels of sHLA-G at the end of treatment between the groups that achieved lesion regression and those that did not (*p* = 0.1670) (Figure 2B). An analysis of paired samples before and after treatment showed that IMQ did not influence sHLA-G expression among those who were successfully treated (*p* = 0.3395) (Figure 2C). Individuals who did not respond to the treatment exhibited high tissue sHLA-G expression upon admission (*p* = 0.0136) (Figure 2D). We did not observe a significant difference in sHLA-G levels in response to IMQ in the patients successfully treated for both CIN 2 (*p* = 0.3081) and CIN 3 (*p* = 0.4711) lesions (Figure 2E). However, in the unsuccessfully treated patients, we observed high levels of sHLA-G before treatment in the CIN 3 (*p* = 0.0101) but not in the CIN2 (*p* = 0.3699) groups (Figure 2F). These findings suggest that tissue immune competency before stimulation with IMQ affects the response to the therapy.

### 3.3. Effect of IMQ Treatment on Cervical PD-1/PD-L1 Levels

Cervical PD-1 levels before the application of IMQ (Figure 3A) or after the completion of treatment (Figure 3B) had no effect on treatment outcome. In a paired analysis, it was found that IMQ did not significantly influence the increase or decrease in PD-1 expression in treatment success (Figure 3C,E) or failure (Figure 3D,F). However, the difference in PD-1 expression among the CIN3 samples following IMQ treatment was significantly higher in the patients who did not respond to the treatment compared to those successfully treated (30.97 vs. 11.61, *p* = 0.0491), despite comparable levels in the pre-treatment samples (23.68 vs. 21.72, *p* = 0.5490).

An association was observed between high levels of PD-L1 in the samples before IMQ application and the regression of the cervical lesion (*p* = 0.0085, Figure 4A), although no such association was identified in the samples after treatment completion (*p* = 0.7969, Figure 4B). A paired analysis of the samples before and after treatment revealed a decrease in PD-L1 levels in those who were successfully treated (*p* = 0.0509, Figure 4C) and an increase in those who failed treatment (*p* = 0.0404, Figure 4D). The PD-L1 expression pattern before treatment was similar in patients with CIN2 or CIN3 (Figure 4E,F).

### 3.4. The Model of Immune Modulation in Cervical Lesions Associated with IMQ Treatment Outcome

A multivariate analysis showed that the presence of sHLA-G in the lesion at high levels reduced the chances of treatment success by 90%, while high levels of PD-L1 increased the chances of treatment success by up to 10 times. Age also affected the explanatory model. Women over 28 years of age exhibited a 5 times greater likelihood of successful treatment. This observation can be attributed to the fact that nine out of the twelve women who did not respond to treatment had CIN3 and were under 28 years of age, while two out of the four with CIN2 were also under 28 years of age (Table 1).

Additionally, the reciprocal influence of sHLA-G/PD-1, sHLA-G/PD-L1, and PD-1/PD-L1 in response to IMQ application was not significant in the case of successful or failed treatment (Figure 5).

### 3.5. The Effect of IMQ Treatment on HPV E7 Oncoprotein Expression in Cervical Lesion

A paired analysis revealed a reduction in E7 expression in cervical biopsies after the completion of treatment in the patients who responded to treatment with lesion regression (Figure 6A,B). The superimposition of nuclear labelling with DAPI in blue and E7 in green demonstrated the interaction of the viral oncoprotein (E7) with the human genome. Conversely, E7 expression remained comparable before and after treatment in those who did not achieve lesion reduction (Figure 6C,D).

## 4. Discussion

Contemporary data suggest that IMQ is more effective than a placebo but less effective than LLETZ in women with HSILs. Our prior study revealed a moderate (51.9%) histological regression rate, in line with similar outcomes reported in other studies (55%) [6]. Emerging research is dedicated to unravelling the significance of the immune microenvironment in IMQ response, as ongoing initiatives work towards the development of immune biomarkers for IMQ [13]. 

In our current investigation of the impact of IMQ treatment on cervical sHLA-G, PD-1, and PD-L1 expression in HSILs, we made notable observations that provide insight into the therapeutic efficacy of IMQ and potential biomarkers.

Our study revealed that, when successful, IMQ treatment did not significantly influence sHLA-G expression in HSILs. Interestingly, elevated sHLA-G levels upon admission were associated with an increased risk of unsuccessful treatment with IMQ. These findings suggest that baseline sHLA-G levels could serve as a predictive factor for treatment outcomes. Additionally, the observed lack of impact on sHLA-G expression by IMQ raises questions about the direct influence on this immune molecule. HLA-G gene variants are associated with a greater constitutional expression of sHLA-G [14], but the influence of genetic factors on the expression of this biomarker is not addressed here. 

It is also established that high-risk HPV induces aberrant HLA-G expression in cervical lesions. This expression can generate inhibitory signals within the cancer microenvironment, enabling tumour cells to evade immune surveillance and exerting influence on tumour progression and metastasis [15]. This underscores its potential as a biomarker for the diagnosis and prognosis of cervical cancer [10,15,16,17,18]. However, the mode of action of IMQ does not seem to impact the expression of HLA-G, as also indicated by other studies [19].

Regarding the PD-1 pathway, we observed that IMQ did not significantly alter the expression of PD-1. Those who were treated successfully reported more vaginal inflammation events during the treatment protocol. T cell infiltrate may be associated with increased tissue PD-1 expression in low-grade lesions [20], and inflammation is related to HPV infection regardless of the degree of lesion [21]. The increase in PD-1 levels in CIN3 lesions that did not respond to treatment coincides with the persistence of viral activity and a decrease in sHLA-G, corroborating the association of inflammation with the development of cancer. 

Elevated tissue PD-L1 expression before treatment was associated with the successful treatment of IMQ, highlighting its potential as a predictive biomarker. Unlike PD-1, the analysis of PD-L1 expression before and after treatment revealed a reduction in expression in cases of successful treatment. On the contrary, PD-L1 levels increased in cases of treatment failure, reaching levels comparable to those in successfully treated cases. These findings regarding PD-1 and PD-L1 imply that IMQ may not be as effective in locally modulating the PD-1 and PD-L1 pathway, potentially influencing immune regulation and the maintenance of an inflammatory state.

The binding of PD-1 to PD-L1 transmits an inhibitory signal that downregulates T cell activation, proliferation, cytotoxic activity, and cytokine production [22]. Exposure to IMQ increased PD-1 expression in keratinocytes; however, in the same study, PD-1 expression was not elevated in lymph nodes [23]. Our findings, combined with those of the aforementioned study, do not conclusively support the notion that IMQ affects PD-1 expression in cervical HSILs. The potential role of combination therapy with an anti-PD-1 antibody, which has shown promise in skin lesions of various cancer types [24], warrants further investigation.

The PD-1/PD-L1 pathway is upregulated in HPV-associated SILs, and this alteration negatively regulates cervical cell-mediated immunity to HPV, thereby contributing to the progression of HPV-related SILs [25]. Results from a study on Langerhans cells (LCs) indicate that IMQ may exert a regulatory effect on PD-L1, inducing PD-L1 expression in LCs in both ear skin and skin-draining lymph nodes [26]. The expression of PD-L1 was reported in the basal layer, where the RNA of the viral E6 and E7 oncoproteins is most abundant [27]. We showed that IMQ treatment failure included the persistence of E7 activity, decreased sHLA-G, and increased PD-L1 without a marked change in PD-1.

The drawback of IMQ treatment lies in the occurrence of side effects, often leading to discontinuation. However, our results indicate that the majority of women, whether using IMQ at least twice per week or irregularly, were successfully treated. Consequently, our findings suggest that the number and duration of side effects did not significantly impact the response to treatment. Further validation through additional studies is warranted to confirm these observations.

Out of the 43 patients, IMQ treatment led to the regression of the cervical lesion in 81% of cases with CIN2, whereas in CIN3, the regression rate was lower at 45.5%. The mechanism determining varying responses to IMQ treatment among patients remains unknown. Genetics could explain it, but there is still no gene profile that answers this question. Another fact to clarify is the antiviral action mechanism of IMQ. While our results demonstrate a reduction in E7 expression, confirmation of a decrease in viral load is not established. In future studies, we aim to include labelling for the oncoprotein E6 and the structural viral protein L1 to assess the impact of IMQ on viral load.

The strength of our study lies in the use of biopsies obtained from a randomised clinical trial, ensuring a thorough analysis. Patient monitoring was conducted carefully, and extensive efforts were made to minimise external factors influencing the cervical microenvironment, aside from IMQ. However, there are limitations to consider. Like in other studies, the clinical response was assessed post-biopsy and IMQ treatment, making it challenging to completely rule out the possibility that the observed clinical responses were solely due to IMQ treatment. Also, the visual analysis of IHC staining in cervical samples, collected before and after IMQ treatment, indicated an unclear association with IMQ treatment, leaving uncertainty about whether another factor could have contributed to the observed outcomes. Despite the challenges, the semiquantitative analysis of molecule expression at the initiation and conclusion of treatment, relative to the severity of the lesion, provided valuable insights. Additionally, the IMQ-treated group included both CIN2 and CIN3 patients, despite the known higher likelihood of spontaneous regression in CIN2. Lastly, the absence of HPV genotyping in our study underscores the need for future investigations to incorporate HPV genotyping and assess HPV DNA methylation status.

## 5. Conclusions

In conclusion, our study provides valuable insights into the molecular dynamics influenced by IMQ treatment in cervical HSILs. The associations identified between the baseline levels of sHLA-G, PD-1, and PD-L1 and treatment outcomes emphasise the potential utility of these molecules as predictive biomarkers. We showed that the antiviral activity of IMQ associated with immunological competence in cervical mucosa with low sHLA-G levels contributes to the success of the treatment. Furthermore, the anti-inflammatory activity of IMQ can favourably control HSILs by regulating the PD-1/PD-L1 pathway. However, the complexities of the immune response and the diverse outcomes warrant additional exploration to clarify the nuanced mechanisms at play in IMQ treatment within the context of cervical lesions.

## Figures and Tables

**Figure 1 cancers-16-01272-f001:**
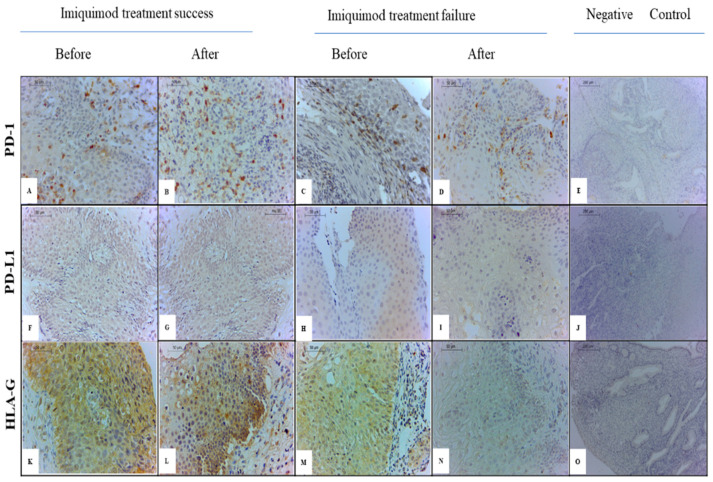
Detection of HLA-G, PD-1, and PD-L1 in cervical mucosa. PD-1 (**A**–**D**), PD-L1 (**F**–**I**), and HLA-G (**L**–**N**) in cervical mucosa before (**A**,**C**,**F**,**H**,**L**,**M**) and after (**B**,**D**,**G**,**I**,**K**,**N**) IMQ treatment. PD-1 labelling in patients with successful treatment (**A**,**B**) and treatment failure (**C**,**D**). PD-L1 labelling in patients with successful treatment (**F**,**G**) and treatment failure (**H**,**I**). HLA-G labelling in patients with successful treatment (**K**,**L**) and treatment failure (**M**,**N**). Panels (**E**,**J**,**O**) represent negative controls for PD-1, PD-L1, and HLA-G, respectively (400× magnification).

**Figure 2 cancers-16-01272-f002:**
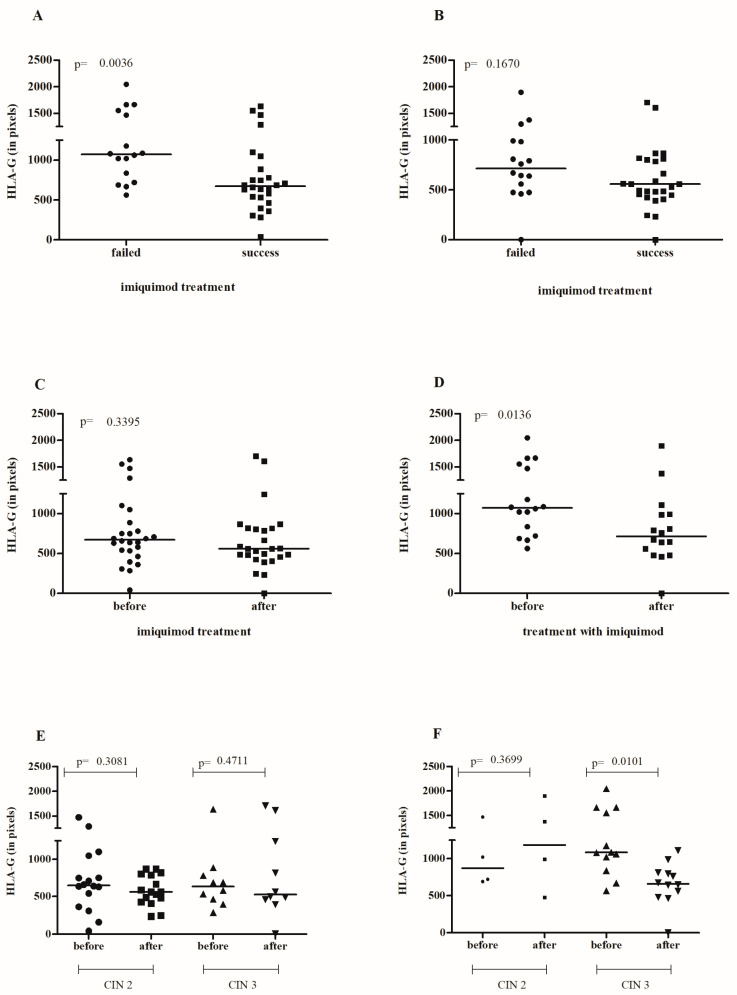
The assessment of sHLA-G staining intensity in cervical lesions before (**A**) and after (**B**) local IMQ therapy in relation to the success of therapy. Paired analysis of pre- and post-treatment sHLA-G levels in patients with successful treatment (**C**) and those with treatment failure (**D**). Evaluation of pre- and post-treatment sHLA-G levels in patients with successful treatment (**E**) and treatment failure (**F**), stratified by the initial diagnosis of CIN 2 or CIN 3.

**Figure 3 cancers-16-01272-f003:**
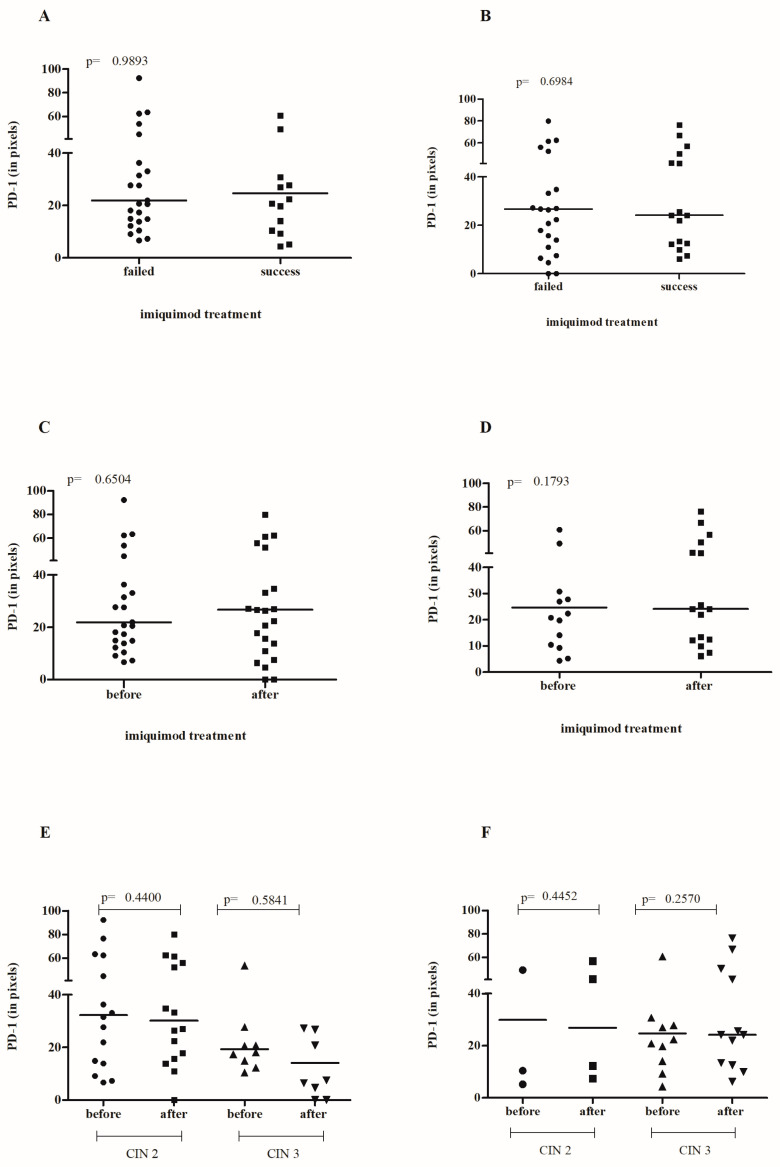
The assessment of PD-1 staining intensity in cervical lesions before (**A**) and after (**B**) local IMQ therapy in relation to the success of therapy. Paired analysis of pre- and post-treatment PD-1 levels in patients with successful treatment (**C**) and those with treatment failure (**D**). Evaluation of pre- and post-treatment PD-1 levels in patients with successful treatment (**E**) and treatment failure (**F**), stratified by the initial diagnosis of CIN 2 or CIN 3.

**Figure 4 cancers-16-01272-f004:**
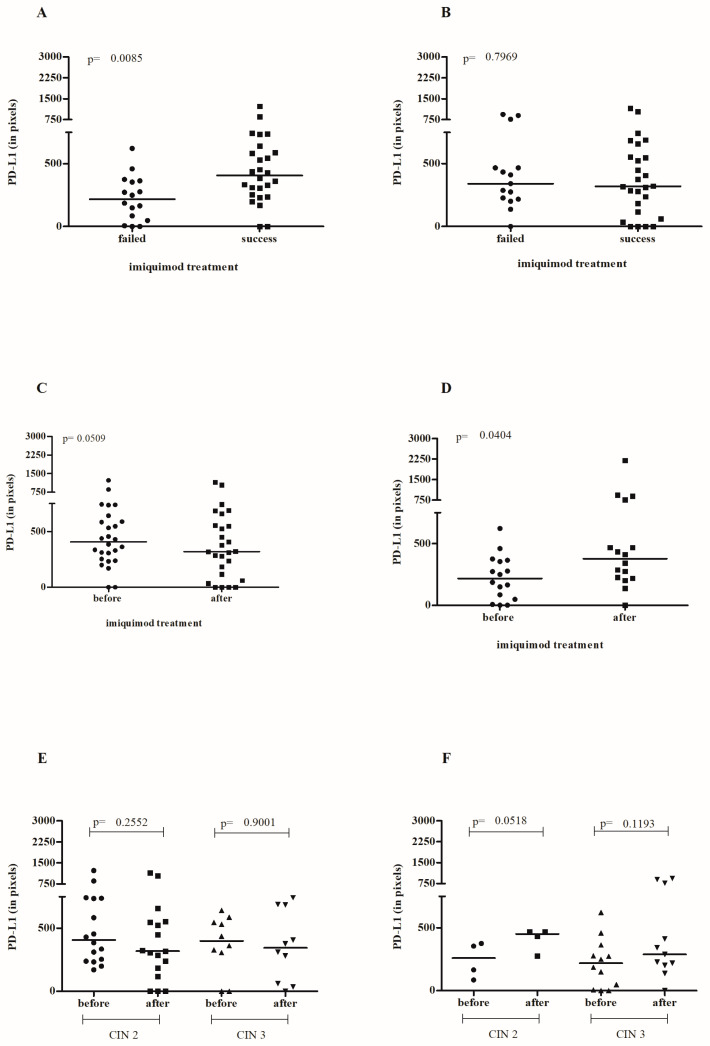
The assessment of PD-L1 staining intensity in cervical lesions before (**A**) and after (**B**) local IMQ therapy in relation to the success of therapy. Paired analysis of pre- and post-treatment PD-L1 levels in patients with successful treatment (**C**) and those with treatment failure (**D**). Evaluation of pre- and post-treatment PD-L1 levels in patients with successful treatment (**E**) and treatment failure (**F**), stratified by the initial diagnosis of CIN 2 or CIN 3.

**Figure 5 cancers-16-01272-f005:**
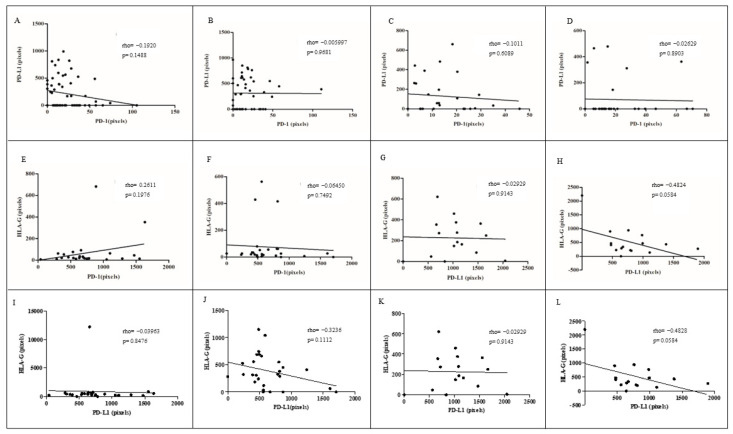
Spearman correlation between HLA-G, PD-1, and PD-L1 expression before and after IMQ treatment. Correlation between PD-1 and PD-L1 expression levels before (**A**) and after (**B**) successful treatment, and before (**C**) and after (**D**) treatment failure. Correlation between HLA-G and PD-1 expression levels before (**E**) and after (**F**) successful treatment, and before (**G**) and after (**H**) treatment failure. Correlation between HLA-G and PD-L1 expression levels before (**I**) and after (**J**) successful treatment, and before (**K**) and after (**L**) treatment failure.

**Figure 6 cancers-16-01272-f006:**
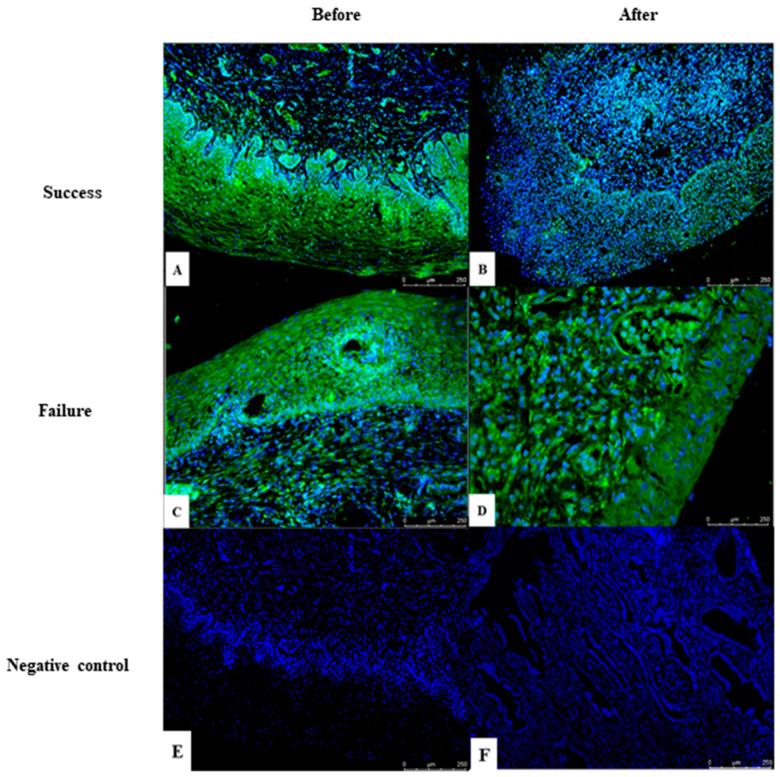
Immunofluorescence of HPV E7 in a cervical biopsy. E7 expression in cervical mucosa with HSIL before (**A**) and after (**B**) successful treatment with topical IMQ. E7 expression in cervical mucosa with HSIL before (**C**) and after (**D**) treatment failure with topical IMQ. Negative control (**E**,**F**) exhibits nuclei stained with DAPI (200× magnification).

**Table 1 cancers-16-01272-t001:** Univariate and multivariate analyses of immune molecules associated with the success of topical imiquimod treatment in cervical lesions. Considering the mean as a cutoff point for HLA-G, age, and the first quartile for PD-L1, sHLAG in the lesion reduces the chances of successful treatment by 86%, while PD-L1 increases the chances of successful treatment by up to 10 times.

Univariate Analysis	Success of Treatment				OR	CI (95%)		*p*-Value
	No		Yes					
	N	%	N	%		Low	Up	
HLAG								
<898,216	5	31.25	20	76.92	1.00			
≥898,216	11	68.75	6	23.08	0.14	0.03	0.52	0.0052
PDL1								
<189,626	8	50.00	3	11.54	1.00			
≥189,626	8	50.00	23	88.46	7.67	1.76	42.23	0.0101
Age								
<28.49	11	68.75	13	48.15	1.00			
≥28.49	5	31.25	14	51.85	2.37	0.67	9.29	0.1931
Multivariate Model								
HLAG								
≥898,216					0.10	0.01	0.48	0.0079
PDL1								
≥189,626					10.11	1.76	84.04	0.0160
Age								
≥28.49					4.90	0.95	35.03	0.0770

## Data Availability

Data are contained within the article.

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
