# Peer review of "Modulation of sHLA-G, PD-1, and PD-L1 Expression in Cervical Lesions Following Imiquimod Treatment and Its Association with Treatment Success"

_cancers, 2024, doi:10.3390/cancers16071272_

Round 1

Reviewer 1 Report

Comments and Suggestions for Authors

                  Comment to Editors and Authors

This article dealt with imiquimod (IQM) treatment for squamous intraepithelial lesions (SIL) being caused from human papilloma virus (HPV) infection. The infection of HPV, especially HPV 16 have E6 protein and E7 protein. E6 inactivate P53 protein of squamous cell, which is major anti-cancer gene product and the latter, E7, conjugate the RB protein, which is another important anti-cancer gene product, then free E2F is released which cause the progression of oncogenes. Both effects were the major oncogenic pathways of cervical cancer.

IQM is agonist of toll like receptor 7 (TOR 7) and the down-stream of immune pathway such as soluble HLA-G which caused positive self-recognition of tumor cell for intrinsic immunity. Therefore, induction by IMQ into HPV-infected cells would facilitate the activation of immune systems of the microenvironments around the tumor cells and exterminate them.

First, in this trial, IMQ cream had been administrate by patient’s maneuvers, so it was unclear the device had put correct location adequately to the cervical lesion or around its vaginal wall lesion. Even by gynecologist, it would be difficult to administered IMQ-cream adequately to the SIL lesion because of hidden covers by secretion or diversity in direction of portio vaginalis. Secondly, her sexual partners are same, newly, or recurrent infection would occur (relapse, or persistent). If those conditions stated above were elucidated and eliminated, this analysis should be more appropriate. These conditions might make the data instability of sHLA-G.

Author should make some comments the conditions stated above.

Despite the unstable data of sHLA-G, the concept is acceptable, and the result of this trial is promising and a milestone for strategy to cervical cancer.

Author Response

Reviewer 1:

This article dealt with imiquimod (IQM) treatment for squamous intraepithelial lesions (SIL) being caused from human papilloma virus (HPV) infection. The infection of HPV, especially HPV 16 have E6 protein and E7 protein. E6 inactivate P53 protein of squamous cell, which is major anti-cancer gene product and the latter, E7, conjugate the RB protein, which is another important anti-cancer gene product, then free E2F is released which cause the progression of oncogenes. Both effects were the major oncogenic pathways of cervical cancer.

IQM is agonist of toll like receptor 7 (TOR 7) and the down-stream of immune pathway such as soluble HLA-G which caused positive self-recognition of tumour cell for intrinsic immunity. Therefore, induction by IMQ into HPV-infected cells would facilitate the activation of immune systems of the microenvironments around the tumour cells and exterminate them.

First, in this trial, IMQ cream had been administrate by patient’s manoeuvres, so it was unclear the device had put correct location adequately to the cervical lesion or around its vaginal wall lesion. Even by gynaecologist, it would be difficult to administered IMQ-cream adequately to the SIL lesion because of hidden covers by secretion or diversity in direction of portio vaginalis. Secondly, her sexual partners are same, newly, or recurrent infection would occur (relapse, or persistent). If those conditions stated above were elucidated and eliminated, this analysis should be more appropriate. These conditions might make the data instability of sHLA-G.

Author should make some comments the conditions stated above.

Despite the unstable data of sHLA-G, the concept is acceptable, and the result of this trial is promising and a milestone for strategy to cervical cancer.

Response to the Reviewer 1.

The risk of inadequate contact between the device containing IMQ and the patient's cervical lesion is inherent in the study design. However, to mitigate this, a menstrual cup was utilized due to its user-friendly nature, ease of use, and monthly familiarity for some patients. It is worth noting that, in the majority of other studies evaluating the effect of IMQ treatment, a vaginal applicator (similar to that used for administering vaginal antibiotics) was employed, which has similar disadvantages to menstrual cup.

The evaluation of the IMQ effect was conducted over a relatively short follow-up period, and the sexual partners were assumed to be consistent. While it would have been intriguing to monitor cervical HPV infection through genotyping, this was not feasible due to the absence of such provisions in the original protocol (mentioned in line 389).

Reviewer 2 Report

Comments and Suggestions for Authors

The authors evaluated the predictive role of response to treatment with imiquimod by tissue expression of some immune checkpoints in CIN.

The study is worthy of publication, but some issues need to be clarified.

Here are my comments:

Abstract 

Please clarify the acronyms also in the abstract the first time you use them.

“The study employed a rigorous schedule of biopsies to assess treatment effectiveness, and statistical analyses were applied to interpret the data. Ethical considerations were adhered to, and the research methods were based on a previously approved clinical trial;” 

These sentences seem superfluous in the abstract and I would delete them. I would describe much better the patient cohort, it is not even reported the number of patients included in the study, a much more important data to highlight in the abstract.

Introduction 

IMQ can be used for other intraepithelial neoplasms such as vulvar paget. I would cite it in the list of pathologies where IMQ can be used for completeness (lines 60-62). DOI: 10.1016/j.ajog.2022.04.012 Also in this introductory context I would cite the ESGO guidelines for the treatment of pre-invasive vulvar lesions (VAIN-VIN) 10.1097/LGT.0000000000000683.doi: 10.1136/ijgc-2021-003262.

Consider this review when you talk about immunotherapy for Cervical cancer DOI: 10.3390/ijms23073559 the reference 8 you cite is on melanoma!!!

Reference 9. It’s from 2015. Surely you can cite some more recent papers on the topic.

Material and methods 

Where did the patients come from? Organized screening? Are there data on HPV genotypes (if performed)? 

Did you exclude patients who were undergoing immunosuppressive or cortiscosteroid based treatment or were immunodepressed? Can you specify?

Lines 157-158: can you specify with what criterion you chose the variables to put in multivariate analysis?

Results

Please start with the total number of patients included. It is more correct to report this data in the results rather than in materials and methods. 

Please check the acronyms well thorugh the whole manuscript: lines 168: CI, OD

Declare in the results the age of the patients, their smoking habitus… you must also declare in material and methods that you have collected these data as clinical information.

How many patients did not complete the treatment due to side effects? Can you also specify the type of any side effects? Could this issue affect the response to therapy?

“Vaginal inflammation observed after ten weeks of treatment occurred…” how did you define vaginal inflammation?

Can you also bring the absolute numbers of responders and non-responders and clearly define the categories? Did you also evaluate any partial responses? And if so, how were they evaluated?

Lines 177-180 what do you mean by “…indicated a potential relationship between changes in 178 sHLA-G and PD-L1, and possibly PD-1 levels with treatment success. However, the association with IMQ treatment remained unclear, and it was uncertain whether another factor could be contributing to the observed outcomes”… it is not clear, in this section limit yourself to exposing results. Comment then in the discussion.

Perhaps in the analyses you should also evaluate the smoking status and CINII vs CINIII

Discussion: Start also with the results so far obtained from the studies that have evaluated the treatment of CIN with IMQ. Consider these: doi:10.1002/ijgo.14953 https://doi.org/10.1002/jmv.29238

Reference 25 is about psoriasis. I would limit myself to discussing works that are based on oncological immunotherapy possibly on cervical cancer.

Other authors have evaluated immunological biomarkers to understand the role of IMQ in CIN: DOI: 10.1136/jitc-2022-005288 comment on the results of other authors as well.

Describe the strengths and limitations of your study.

CIN II can also regress spontaneously, this issue should be considered in the discussion.

Comments on the Quality of English Language

In shorts:

  • Introduction and discussion need to be revised, outdated or unsuitable references have been used or omitted some important ones (e.g. ESGO guidelines, important recent reviews and/or systematic reviews)
  • some formal and conceptual methodological aspects need to be deepened and explained.

Author Response

We thank the Reviewer 2 for his suggestions that help to improve the manuscript. Responses to each comment are below.

OK- Comment 1. Abstract 

Please clarify the acronyms also in the abstract the first time you use them.

“The study employed a rigorous schedule of biopsies to assess treatment effectiveness, and statistical analyses were applied to interpret the data. Ethical considerations were adhered to, and the research methods were based on a previously approved clinical trial;” 

These sentences seem superfluous in the abstract and I would delete them. I would describe much better the patient cohort, it is not even reported the number of patients included in the study, a much more important data to highlight in the abstract.

  1. Response to the comments in Abstract: We edited the abstract to clarify the acronyms and highlight the patient data and results.

Abstract: (1) Background: Cervical intraepithelial neoplasia (CIN) is a precancerous condition linked to human papillomavirus (HPV infection), often necessitating surgical interventions carrying the risk of preterm births. This study explores the potential of imiquimod (IMQ), as a non-invasive alternative treatment. The focus is on understanding IMQs impact on immune checkpoint molecules, particularly PD-1, PD-L1, and sHLA-G, which play pivotal roles in shaping immune responses and cancer progression.

(2) Methods: Forty-three patients diagnosed with high-risk squamous intraepithelial lesion (HSIL, p16 positive) self-applied 5% imiquimod (IMQ) encapsulated in sachets containing 250 grams of cream into the vaginal cavity three times a week for 16 weeks. The impact of IMQ therapy on cervical lesion regression was assessed through immunohistochemistry, examining changes in sHLA-G, PD-L1, and PD-1 levels. The antiviral activity of IMQ was evaluated through HPV-E7 immunofluorescence.

(3) Results: IMQ treatment demonstrated efficacy, leading to lesion regression. sHLA-G levels in CIN before starting IMQ application were associated with unsuccessful treatment (P= 0.0036). IMQ did not significantly alter the expression of the PD-1. We observed a decrease in PD-L1 levels in those who were successfully treated (P = 0.0509), and a reduction in HPV burden.

OK- Comment 2. Introduction 

IMQ can be used for other intraepithelial neoplasms such as vulvar paget. I would cite it in the list of pathologies where IMQ can be used for completeness (lines 60-62). DOI: 10.1016/j.ajog.2022.04.012 Also in this introductory context I would cite the ESGO guidelines for the treatment of pre-invasive vulvar lesions (VAIN-VIN) 10.1097/LGT.0000000000000683.doi: 10.1136/ijgc-2021-003262.

Consider this review when you talk about immunotherapy for Cervical cancer DOI: 10.3390/ijms23073559 the reference 8 you cite is on melanoma!!!

Reference 9. It’s from 2015. Surely you can cite some more recent papers on the topic.

  1. Response to the comments in Introduction: We edited the introduction to include the reference DOI: 10.1016/j.ajog.2022.04.012, the Paget trial and the use of IMQ. However, we did not include the reference DOI: 10.1136/ijgc-2021-003262, since it is specific for vulvar lesions. 

We corrected the citation of reference 8 as suggested and also in reference 15.

Former Ref. 8: Turinetto M, Valsecchi AA, Tuninetti V, Scotto G, Borella F, Valabrega G. Immunotherapy for Cervical Cancer: Are We Ready for Prime Time? International Journal of Molecular Sciences. 2022; 23(7):3559. https://doi.org/10.3390/ijms23073559

Former Ref. 9: Guo,X.,Zhang,J.,Shang,J.,Cheng,Y.,Tian,S.,& Yao,Y.(2023).Human leukocyte antigen-G in gynaecological tumours. International Journal of Immunogenetics, 50, 163–176. https://doi.org/10.1111/iji.12626

We corrected and updated the references. 

OK -Comment 3. Material and methods 

Where did the patients come from? Organized screening? Are there data on HPV genotypes (if performed)? 

We corrected the text in lines 92-94. Genotyping was unfortunately not performed. We commented on that in the discussion.

Did you exclude patients who were undergoing immunosuppressive or cortiscosteroid based treatment or were immunodepressed? Can you specify?

We corrected the text in lines 92-94. 

(Patients enrolled in the trial were sourced from organised screening and were not immunodeficient, immunocompromised, or undergoing corticosteroid treatment.)

Lines 157-158: can you specify with what criterion you chose the variables to put in multivariate analysis? 

  1. Response to the comments in Material and Methods

We summarized the methods from the original study (ClinicalTrials.gov Identifier: NCT04859361) cited in reference 11. We added 

We also edited the lines 157-158 (now 163-165) for completeness, as follows: The variables with values of p<0.025 were selected and ordered from the lowest to the highest significance value and inserted into the model, leaving only the variables that presented a significance of p<0.05.

Comment 4. Results

  1. Response to the comments in Results below each comment

Please start with the total number of patients included. It is more correct to report this data in the results rather than in materials and methods. 

  • Corrected in line 175. 

Please check the acronyms well through the whole manuscript: lines 168: CI, OD

  • Corrected in lines 168 and 177.

Declare in the results the age of the patients, their smoking habitus… you must also declare in material and methods that you have collected these data as clinical information.

  • Corrected in lines 99-100.

How many patients did not complete the treatment due to side effects? Can you also specify the type of any side effects? Could this issue affect the response to therapy?

  • Added results in lines 188-198. Also, we added a paragraph in the discussion regarding side effects and success of treatment (lines 359-346).

“Vaginal inflammation observed after ten weeks of treatment occurred…” how did you define vaginal inflammation?

  • Corrected in line 198. Vaginal inflammation, assessed through a clinical examination with or without a bacterial swab after 10 weeks of treatment with IMQ, occurred in a similar proportion among women who exhibited lesion regression (41% of 27) and those who did not (50% of 16) at the completion of the treatment protocol (P= 0.7846).

Can you also bring the absolute numbers of responders and non-responders and clearly define the categories? Did you also evaluate any partial responses? And if so, how were they evaluated?

  • Hopefully this was clarified in our previous response in the results (188-198).

Lines 177-180 what do you mean by “…indicated a potential relationship between changes in 178 sHLA-G and PD-L1, and possibly PD-1 levels with treatment success. However, the association with IMQ treatment remained unclear, and it was uncertain whether another factor could be contributing to the observed outcomes”… it is not clear, in this section limit yourself to exposing results. Comment then in the discussion.

  • We changed the text and transferred the deleted sentences to discussion. 

Perhaps in the analyses you should also evaluate the smoking status and CINII vs CINIII. 

  • Baseline characteristics were presented in the original study (reference 11) and smoking status was similar between groups. 

Comment 5. Discussion

  1. Response to the comments in Discussion

Discussion: Start also with the results so far obtained from the studies that have evaluated the treatment of CIN with IMQ. 

  • The study from Consider these: doi:10.1002/ijgo.14953 https://doi.org/10.1002/jmv.29238

We have included the text and a reference from Van de Sande (line 303-306).

Reference 25 is about psoriasis. I would limit myself to discussing works that are based on oncological immunotherapy possibly on cervical cancer.

There are studies about immunological markers in cervical (pre)cancer, but they are designed differently. In this study we wished to remain focused on IMQ, so we did not explore other immunotherapeutic options. There is very little data about IMQ, cervix and PD-1 and PD-L1, so if it is possible we would like to keep this reference. 

Other authors have evaluated immunological biomarkers to understand the role of IMQ in CIN: DOI: 10.1136/jitc-2022-005288 comment on the results of other authors as well.

Added in the discussion section (line 306-308).

Describe the strengths and limitations of your study.

Added in lines 374-384

CIN II can also regress spontaneously, this issue should be considered in the discussion.

Incorporated in the limitations of the study (line 381).

OK- Comment 6. Comments on the Quality of English Language

In shorts:

Introduction and discussion need to be revised, outdated or unsuitable references have been used or omitted some important ones (e.g. ESGO guidelines, important recent reviews and/or systematic reviews)

some formal and conceptual methodological aspects need to be deepened and explained.

  1. Response to the comments in Material and Methods

The authors reviewed the manuscript, editing as suggested by Reviewer 1, special attention was given to citing references. Regarding questions about the clinical trial, the clinical trial itself was previously published and endorsed in the methodology (Former reference 11). In this manuscript, mucosal immunity was further explored in the face of the intervention described in detail previously. This aspect was better clarified in the Methods section in the new version of the manuscript.

Round 2

Reviewer 2 Report

Comments and Suggestions for Authors

This referee's comments have been clarified